# Aerial Spraying and Its Impacts on Human Health in Banana-Growing Areas of Ecuador

**DOI:** 10.3390/healthcare12202052

**Published:** 2024-10-16

**Authors:** Mauricio Guillen, Juan Calderon, Freddy Espinoza, Lizan Ayol

**Affiliations:** School of Health Sciences, Universidad Estatal de Milagro, Milagro 091050, Ecuador; jcalderonc@unemi.edu.ec (J.C.); fespinozac@unemi.edu.ec (F.E.); layolp@unemi.edu.ec (L.A.)

**Keywords:** aerial fumigation, banana industry, cholinesterase, gastrointestinal, neurological, reproductive and dermatological disorders

## Abstract

The present work examines the relationship between aerial spraying and its health impacts on the population living in the banana production areas of Ecuador (the rural sectors of the cantons Milagro and Naranjito, Guayas Province). **Objectives**: the objectives of this study are to obtain information on sanitation, basic services, and environmental rationality and to interpret the low levels of cholinesterase and prevalent diseases among the population. **Methods:** the methodology involved a face-to-face questionnaire, the formal authorization of an informed consent document, and venipuncture for cholinesterase tests. The information was processed in the EPI–INFO system 7.2 (statistical software for professionals and researchers dedicated to public health), with the certification of protocols issued by the Bioethics Committee of the Kennedy Hospital Clinic of Ecuador. **Results:** the results showed that 89.5% of inhabitants do not have access to drinking water, 92.5% do not have a sewage disposal service, 97.50% experience aerial spraying at their homes or workplaces, and 57% have low cholinesterase levels. Additionally, several gastrointestinal, respiratory, neurological, dermatological, and reproductive disorders were detected among the inhabitants. **Conclusions**: we found that companies in the banana sector have not implemented corporate social responsibility measures. For example, no blood tests are conducted to monitor cholinesterase levels or to track hereditary disorders. Moreover, entities such as the Ministry of Public Health have not taken action to serve this at-risk population.

## 1. Introduction

As a banana-exporting country, Ecuador has a long history of banana management. This management helps to balance trade and payments, enabling the country to become one of the largest banana exporters worldwide. However, banana management includes aerial spraying to eliminate pests that affect bananas, increasing pesticide use and their potential neurotoxic effects on human health and the environment [1]. Although aerial spraying leads to environmental contamination, currently there is no safe drinking water plan in Ecuador’s rural areas. Spraying also pollutes the air and soil of the agricultural environments in question.

Approximately 25 million agricultural workers worldwide experience unintentional pesticide poisoning each year [2]. The key problem in banana management is the use of traditional pesticides, i.e., fumigation using chemicals, rather than natural or organic mixtures that perform the same function against banana pests. Traditional fumigation is more cost-effective in terms of both time and financial resources but has a very high human cost. The lack of detailed information on safety protocols, access to protective equipment, and agricultural practices must also be considered.

Residents in proximity to the banana-farming areas, including those who are also workers (responsible for land preparation and the cultivation and harvesting of bananas), are exposed to daily fumigation, without a defined frequency or set schedule for herbicide application. Prolonged exposure to chemicals poses significant health risks, potentially leading to severe and life-threatening conditions, including allergic reactions, gastrointestinal, dermatological, respiratory, neurological, reproductive, and ocular disorders, as well as an increased risk of certain types of cancer [3].

The majority of workers in these plantations belong to banana-growing areas in the cantons of Milagro and Naranjito, with limited access to health resources and low purchasing power [4]. In addition, legislation on the use of pesticides is often limited or unenforced, resulting in poorly controlled agricultural practices and increased risks to human health and water quality in the surrounding rivers [5]. The lack of effective protection policies has exacerbated these problems [6].

Previous studies have predominantly focused on individuals working in the agricultural sector in various localities, as they handle chemicals for fumigation. However, over the past few years, there has been a lack of studies examining the long-term health impacts on residents who are living in these agricultural environments. In this area, there is a noticeable absence of medical facilities or social responsibility foundations to help monitor cholinesterase levels in the population. Cholinesterase levels, measured through blood tests, serve as an indicator of pesticide exposure due to the continual inhalation of agrochemicals. Reduced cholinesterase levels are associated with increased vulnerability to respiratory and dermatological diseases, among other health complications [7].

The findings of this research are not only relevant at the local level in Ecuador but could also contribute to global scientific knowledge on pesticide exposure in agriculture and its implications for human and environmental health in rural agricultural areas more broadly [8].

This observational analysis aims to evaluate the living conditions of residents concerning sanitation, access to basic services, environmental sustainability, cholinesterase levels, and the incidence of disease. Monitoring cholinesterase levels is essential for safeguarding public health, fostering sustainable agricultural practices, and informing policies that balance economic growth with the protection of human health and environmental integrity.

## 2. Methods

### 2.1. Study Design

The present study applied a mixed research approach with a descriptive–correlational observational scope (based on the evaluation of blood samples). Data and sample collection took place in March 2023, with the voluntary participation of 200 inhabitants from the communities of Venecia Central, Córdoba, Vuelta del Piano, and Alegría, which are banana-growing areas in the cantons of Milagro and Naranjito, Ecuador. The inclusion criteria required all participants to be adults (over 18 years old), reside in the study area, be in good health, and be willing to understand and sign the informed consent. Exclusion criteria included pregnant women, individuals with disabilities, minors under 18 years of age, and participants who, due to religious beliefs, did not wish to provide blood samples. No mathematical formula was used to generate the sample, as many individuals were unwilling to participate due to the blood sample collection process; therefore, a voluntary sample was applied. All participants signed the informed consent and answered a physical questionnaire with questions related to access to basic services and medical care, knowledge of their environmental reality, and their current illnesses for subsequent analysis. For the venipuncture, nurses who are associate researchers participated. They applied alcohol and an elastic band around the upper part of the participant’s arm to apply pressure to the area, allowing the vein underneath to fill with blood. A needle was then inserted into the vein, and the blood was collected in a sealed container.

### 2.2. Laboratory Analysis of Plasma Cholinesterase

Advanced analytical techniques were used to identify and quantify cholinesterase levels in the study subjects. Once the respondents signed their informed consent, the staff of the National Reference Center of Toxicology of INSPI-Guayaquil, Ecuador (National Institute of Public Health Research) and Pazmiño Laboratory-Milagro, Ecuador worked in the area to collect the biological samples required to analyze cholinesterase quantitatively [9]. The samples were then transported to the laboratories in coolers containing ice packs. The samples were kept at 2–8 °C during transportation.

Plasma samples were utilized to determine cholinesterase activity levels. The samples were aliquoted into 250 µL volumes and stored at room temperature (20 °C) or refrigerated at +4 °C for up to one month. The analyses were conducted using Liquiform cholinesterase kits from Labtest Diagnóstica S.A., following the Ellman method. This technique quantifies thiocholine production, which is proportional to the cholinesterase enzyme activity in the blood samples. Higher cholinesterase activity correlates with increased thiocholine production. Whole blood samples were placed in red bags, labeled, and stored for intermediate delivery to the environmental manager. For external management, the environmental manager endorsed by the Ministry of Environment and contracted by INSPI collects waste for final disposal. These procedures are similar to those used in other countries [10].

The parameters for the plasma cholinesterase level analysis were as follows:Units: = units/liter (U/L) Reference Values = 4659 − 144,432

This research was approved by the Bioethics Committee of the Kennedy Hospital Clinic (Project ID HCK-CEISH-2023-002), which is authorized by the Ministry of Public Health of Ecuador to approve all protocols governing informed consent, collection processes, the transportation and storage of serological samples, etc. 

### 2.3. Data Analysis

Once the fieldwork stages were completed, all information was entered into the statistical programs R (General Public License (GPL), IBM SPSS Statistics—Statistical Package for the Social Sciences (SPSS, version 28.0), and Epi Info (Statistical System for Health). Information on the basic services and health of the study population was analyzed along with their access to public health, contamination of their food and drinking water, cholinesterase levels, and prevalent diseases were anticipated to be significant factors. Prevalence tests were carried out to establish the relationships between variables, such as low cholinesterase levels and their relationships with certain health disorders [11]. To interpret the results, we also applied a relative value analysis and odds ratio to associate the variables and enable us to determine a higher or lower risk of low cholinesterase levels.

## 3. Results

Using the aforementioned statistical programs, we obtained information from the project beneficiaries regarding access to basic services such as external electricity supply, hospital care, affiliation with the Instituto Ecuatoriano de Seguridad Social (IESS), an autonomous public insurance institution responsible for providing social security services and benefits, knowledge of the health risks due to their environmental condition under spraying, their current illnesses, and subsequently plasma cholinesterase levels [12]. The questionnaire was applied to the 200 participants, and blood samples were taken regardless of whether or not the individuals worked on a banana plantation. However, employment notwithstanding, all participants lived in an environment close to a banana plantation (Table 1).

Firstly, the activities of the inhabitants in the rural sectors were analyzed, including their current employment status. In total, 61.5% reported that they work on banana plantations in the area. The remaining percentage do not work on banana plantations but live in the same environment subject to aerial spraying and must be monitored for cholinesterase [13]. Notably, only 40.5% of the inhabitants have social security, and only 17% have access to medical care in public hospitals [14].

In total, 89.5% do not have access to potable drinking water, and 92.5% do not have a sewage disposal service. Therefore, sanitation in this area is precarious, which is aggravated by the ongoing fumigation, which affects the water consumed by people and animals. However, 95% of respondents have electricity, and 58.5% have internet service. Consequently, secondary services are commonplace in the study area, whereas basic needs such as potable drinking water and sewage disposal have been neglected, despite being necessary to ensure a stable public health system.

As shown in Table 2, 97.50% of the respondents stated that they currently experience aerial spraying due to banana activity in the area, and 55.50% are aware of the damage to health that can be caused by these spraying activities or have seen the effects on human health among their own families or neighbors. In total, 86.50% are aware that fumigation activities that occur at any time of the day are affecting their food, 83.50% are aware of such effects in the water they consume, 93.50% are aware of effects in the air, and 86% are aware of impacts in the soil of their homes [15]. It is important to highlight that 59% of the respondents, regardless of gender, state that fumigations can cause harm to pregnant women and live births, based on experiences within some of their family members.

The results for cholinesterase activity are as follows:

According to the plasma cholinesterase analysis of all the voluntary participants, 115 people, including 67 women and 48 men, have low plasma cholinesterase levels. This means that 57.5% of the population in the area is at health risk.

Organochlorines enter an organism via digestion and through the skin and have high liposolubility. When organochlorines enter an organism, they invade fatty tissues. Since they cannot be metabolized by the catabolic pathway, they are deposited in fat-rich organs such as the adrenal gland and disrupt hormone synthesis or other functions.

The information in Table 3 shows the parameters used to measure cholinesterase and the total number of participants with low cholinesterase. In total, 115 respondents were found to have low cholinesterase, representing 57.5% of the sample population, while 85 had normal cholinesterase levels. These results indicate an at-risk population that inhales pesticides due to living near areas with aerial spraying. The following section provides an analysis of the results in Table 3 and Table 4 on the relationship between people with low cholinesterase and their current health disorders.

Low cholinesterase levels and respiratory disorders in the population.

Of the 115 people with low cholinesterase levels, 33 women and 12 men reported experiencing respiratory problems such as constant coughing and bronchial damage due to inhaling pesticides from aerial spraying, with a prevalence of 39%, a relative risk of 1.729, an association greater than 1, and an odds ratio of 2.199. This result supports the probability of a respiratory cause-and-effect relationship [16].

Low cholinesterase level and gastrointestinal disorders in the population.

Among the 115 participants with low cholinesterase, 46 women and 30 men noted that they have gastrointestinal disorders due to fumigations, with a prevalence ratio of 0.66087 (66%). The relative risk has a strength of association greater than 1, i.e., of 1.10144, which is sufficient to establish a cause-and-effect relationship between variables. With an odds ratio of 1.299, there is a probability that people with low cholinesterase will experience a gastrointestinal disorder more than once, suggesting the need for a preventive parameter.

Low cholinesterase level and neurological disorders in the population.

Of the 115 participants with low cholinesterase, 19 women and 26 men reported having neurological disorders such as the inability to walk or events involving thrombosis or tremors in the body. We obtained a prevalence of 0.39 (39%) for this relationship. The relative risk corresponds to a strength of association greater than 1, i.e., of 1.10144, which is sufficient to establish minimal cause-and-effect relationships between the variables. Given the odds ratio of 1.099, there is a limited probability that those with low cholinesterase will develop a neurological disorder, suggesting the need for a preventive parameter [17].

Low cholinesterase level and reproductive disorders in the population.

Of the 115 respondents with low cholinesterase levels, 43 women and 9 men reported experiencing reproductive disorders. For women, this included unexpected abortions, children with some types of anomalies or disabilities, and deceased newborns. For men, it included issues with infertility and having children with disabilities. The inhabitants associated all of these ailments with the fumigations. However, the Epi Info table indicates a prevalence of 45% with a relative risk of 1.325, which reflects an association greater than 1 and an odds ratio of 1.5939. This result establishes a potential probability that those with low cholinesterase will develop reproductive disorders more than once when ingesting pesticides from the environment or contaminated water [18].

Low cholinesterase levels and dermatological disorders in the population

For this parameter, we found that 31 women and 42 men of the 115 participants with low cholinesterase levels suffer from dermatological problems such as skin blemishes, which are generally permanent. Again, the respondents reported that these blemishes are the products of aerial spraying and have, on several occasions, fallen onto their skin—primarily the arms and face. Here, we obtained a prevalence of 63%, a relative risk of 1.3005, and an odds ratio of 1.823. These indicators reflect a considerable association between the variables under study. These issues are commonplace because banana companies do not have a planned spraying schedule.

## 4. Discussion

The results indicate several negative health factors. As previously noted, those living in the study areas represent an at-risk population because they do not have optimal drinking water or sewage disposal systems. This situation is aggravated by the aerial spraying in the area. These pesticides fall on water wells and natural reservoirs used for drinking water, contaminating them and endangering the health of humans and domestic animals. Most of the participants recognize that pregnant women and newborns with disabilities face problems due to pesticide use. The field research and blood sampling revealed families containing members with disabilities. This problem is so common that many parents did not want to take the serological tests because such tests had already been conducted, and the Ministry of Public Health had forbidden them to have more offspring. These participants claim that this reality is due to chemical fumigation and noted that other families face the same problems. 

Farmers exposed to pesticides regularly should be tracked. Organophosphates and carbamates enter an organism through respiration, ingestion, and skin, inhibiting the activity of the cholinesterase enzyme in the transmission of nervous impulses in the autonomic nervous system and causing cholinergic muscarinic manifestations such as excessive salivation, lacrimation, excessive urination, diarrhea, vomiting (emesis), bronchorrhea, rhinorrhea, bronchospasm, bradycardia, and miosis. Organophosphates and carbamates also cause nicotinic cholinergic symptoms such as fasciculations, muscular weakness, mydriasis, tachycardia, abdominal pain, and sweating (diaphoresis).

The true strength of this study is that it was able to correlate actual exposure (verified by plasma cholinesterase) with real health problems in the population under fumigation. Additionally, it sparked the participants’ interest in learning about laboratory tests that could alert them to potential health risks [19]. A limitation of this study was the inability to conduct plasma cholinesterase tests on children and pregnant women, due to their status as vulnerable groups. Living in this geographic area under fumigation likely means that a significant percentage of these individuals also have low cholinesterase levels, which could put them at risk. The lack of erythrocyte cholinesterase measurements is also considered a limitation, as this is generally indicative of a real toxic effect when activity is reduced.

## 5. Conclusions

In the study area, the majority of residents lack access to public hospitals and social security, thus having limited access to medical care.

Almost all of them do not have sanitation services or access to potable water and consume and use water from reservoirs and small rivers exposed to aerial spraying. These factors put the population at risk, as aerial pesticides are sprayed throughout the year.

According to the results, nearly half of the inhabitants are aware of the potential health risks posed by fumigation. However, many were unfamiliar with the plasma cholinesterase test and its role in detecting pesticide exposure and related health issues. Various diseases, including gastrointestinal, respiratory, neurological, reproductive, dermatological, and others, are prevalent. Yet, it remains unclear whether these conditions are directly linked to low cholinesterase levels.

Aerial fumigations and their harmful effects on human health and the environment represent a lack of effective corporate social responsibility management within the banana agricultural sector and by local government authorities. As a result, the quality of life for a significant portion of the population is diminished, the incidence of potential disabilities increases, and the life expectancy of area residents is reduced.

## Figures and Tables

**Table 1 healthcare-12-02052-t001:** Analysis of access to basic services and health.

Current Living Conditions	Participants	Percentage
people over 18 years old	**200**	**%**
**Works at a Banana plantation**		
Yes	123	61.5
No	77	38.5
		**100%**
**Currently has Social Security Instituto Ecuatoriano de Seguridad Social**		
Yes	81	40.5
No	119	59.5
**Has access to health care in public hospitals**		
Yes	34	17
No	166	83
**The following basic services are available at home**		
**Drinking Water**		
Yes	21	10.5
No	179	89.5
**Sewage Service**		
Yes	15	7.5
No	185	92.5
**Electric Power**		
Yes	190	95
No	10	5
**Internet Service**		
Yes	117	58.5
No	83	41.5

Source: EPI INFO SPSS.

**Table 2 healthcare-12-02052-t002:** Correspondence analysis.

Current Living Conditions	Participants	Percentage
people over 18 years old	**200**	**%**
**Aerial spraying in nearby plantations**		
Yes	195	97.50
No	5	2.50
		**100**
**Are you aware of the damage to human health caused by fumigation chemicals?**		
Yes	111	55.50
No	89	59.5
**Aerial Fumigation and Contamination in Banana Environments**		
**You and your family have contamination in your food products**		
Yes	173	86.50
No	27	13.50
**You and your family have contamination in your drinking water.**		
Yes	167	83.50
No	33	16.50
**You and your family have air pollution in the environment.**		
Yes	187	93.50
No	13	6.50
**You and your family have soil contamination in your environment.**		
Yes	172	86.00
No	28	14.00
**Do you know the effects of aerial spraying on pregnant women and live births?**		
Yes	118	59.00
No	82	41.00

Source: EPI INFO SPSS.

**Table 3 healthcare-12-02052-t003:** Population at risk and plasma cholinesterase levels.

Low Plasma Cholinesterase Levels with Less than 4659 U/L in the Blood		Canton Milagro	%	Women 58%	Men 42%
		**(population)**			
	Less than 4.659 U/L	**115**	**57.5%**	**67**	**48**
	Between 4.659 and 14.443 U/L	**84**	**42.0%**	**49**	**35**
	Greater than 14.443 U/L	**1**	**5.0%**	**1**	**0**
Total		**200**	**100.0%**		

Source: EPI INFO SPSS.

**Table 4 healthcare-12-02052-t004:** Low cholinesterase level and disease in the population.

Cholinesterase Level	Respiratory Disorder (33 Women and 12 Men)	No Respiratory Disorder	Total	Prevalence	Odds Ratio	Relative Risk
** * low cholinesterase * **	**45**	**70**	**115**	**0.3913**	**2.1997**	1.72997
normal cholinesterase	19	65	84	0.2261		
**Cholinesterase Level**	**Gastrointestinal Disorder** **(46 Women and 30 Men)**	**No** **Gastrointestinal Disorder**	**Total**	**Prevalence**	**Odds Ratio**	**Relative Risk**
* low cholinesterase *	76	39	115	0.66087	1.2991	1.10144
normal cholinesterase	51	34	85	0.6		
**Cholinesterase Level**	**Neurological Disorder** **(19 Women and 26 Men)**	**No Neurological Disorder**	**Total**	**Prevalence**	**Odds Ratio**	**Relative Risk**
* low cholinesterase *	45	70	115	0.3913043	1.0990	1.072931
normal cholinesterase	31	53	85	0.36470588		
**Cholinesterase Level**	**Reproductive Disorder** **(43 Women and 9 Men)**	**No Reproductive Disorder**	**Total**	**Prevalence**	**Odds Ratio**	**Relative Risk**
* low * * cholinesterase *	52	63	115	0.4521739	1.5939	1.325337
normal cholinesterase	29	56	85	0.34117647		
**Cholinesterase Level**	**Dermatological Disorder** **(31 Women and 42 Men)**	**No** **Dermatological Disorder**	**Total**	**Prevalence**	**Odds Ratio**	**Relative Risk**
* low cholinesterase *	73	42	115	0.634782	1.8239	1.30053
normal cholinesterase	41	43	84	0.48809524		

Source: EPI INFO SPSS.

## Data Availability

Data are contained within the article.

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
