# Peer review of "Aerial Spraying and Its Impacts on Human Health in Banana-Growing Areas of Ecuador"

_healthcare, 2024, doi:10.3390/healthcare12202052_

Round 1
Reviewer 1 Report
Comments and Suggestions for Authors
1. Abstract: How does the study objective in the abstract differ from the one at the end of the introduction? Please explain the purpose of the study.
2. Introduction. Is there any statistical data available locally, regionally, or globally on the diseases caused by exposure to this cholinesterase? Please include it in the introduction.
3. Introduction: How does this study differ from previous ones? What gaps does this study seek to fill?
4. Introduction: Explain why this study needs to be conducted in this location. Why not other Ecuadorian locations with high cholinesterase levels?
5. Introduction. How does the cholinesterase process impact workers and the community? Is it a problem at work or with waste management? Please clarify in the introduction.
6. Methods: It is unclear when the data was collected. How were the blood samples collected from the participants?
7. Methods: How many actual populations participated in this study is unclear. What number of samples were taken? What inclusion and exclusion criteria were applied? What type of sampling formula was used?
8. Result: In the table, why are "RISK FACTORS" and "TOTAL" typed C.A.P.L.O.C.K.? Why not other words?
9. Discussion: Why is the number 1 written before the discussion? Please check the spelling of the words.
10. Discussion: "LOW CHOLINESTERASE AND GASTROINTESTINAL DISORDERS IN THE POPULATION." Please enter it correctly. Do you need to use C.A.P.L.O.C.K. in this section? Please format it as justified! This also applies to "LOW CHOLINESTERASE AND NEUROLOGICAL DISORDERS IN THE POPULATION" and other sections!
11. Conclusions: Why did it not answer the purpose of this study?
12. Bibliography. Please type neatly. Pay attention to the spacing and writing format in the guide!
Comments on the Quality of English LanguageModerate editing of the English language needed
Author Response
Good afternoom, esteemed reviewers.
It is a pleasure to greet you. I would like to express my gratitude for the comments made on this manuscript. I would like to inform you that we have attached a document explaining the corrections made to the manuscript point by point. Additionally, comments have been added in the article itself to each paragraph based on your observations and the changes made.
Thank you very much for your attention.
Observation |
Correction |
1. Abstract: How does the study's objective in the abstract differ from the one at the end of the introduction? Explain the purpose of the study. |
Pag 2. We have written the purpose of the study in both the abstract and the introduction, referring to conditions of basic services, environmental awareness, cholinesterase levels in plasma, and prevalent diseases in the area. |
2. Introduction: Are there statistical data available at the local, regional, or global level on diseases caused by exposure to this cholinesterase? Include this data in the introduction. |
Page 2. We found statistical data for this point, which can be seen in the comments in the document. |
3. Introduction: How does this study differ from previous ones? What gaps does this study aim to fill? |
Pages 1 and 2. No previous studies have been conducted in this setting. The difference is that in this area, plasma cholinesterase tests have never been carried out on the permanently residing population to establish health risks. It is necessary to inform the residents about these laboratory tests so that hospitals can assist them. |
4. Introduction: Explain why it is necessary to carry out this study in this location. Why not in other places in Ecuador with high cholinesterase levels? |
Page 1. Perhaps in some banana plantations in Ecuador, plasma cholinesterase tests have been conducted, but only on the workers, not on the population living near the fumigation areas. |
5. Introduction: How does the cholinesterase process impact workers and the community? Is it a work or waste management issue? Clarify in the introduction. |
Page 2. We have written about the harm caused by low cholinesterase levels to people who are constantly in contact with or inhaling pesticides. |
6. Methods: It is unclear when the data were collected. How were the blood samples of the participants collected? |
Page 3. The collection date, the total number of samples, and the process followed by nurses for blood extraction are mentioned. |
7. Methods: It is unclear how many real populations participated in this study. How many samples were taken? What inclusion and exclusion criteria were applied? What type of sampling formula was used? |
Page 3. The real populations, such as Venecia Central, etc., the number of samples, and the inclusion criteria (e.g., being of legal age, a resident of the area) and exclusion criteria (e.g., pregnant women, minors) are mentioned. |
8. Results: In the table, why are "RISK FACTORS" and "TOTAL" written in uppercase? Why not other words? |
Page 6. The words have been changed, with the first letter capitalized and the rest in lowercase. |
9. Discussion: Why is the number 1 written before the discussion? Please check the spelling of the words. |
Page 7. The number 1 has been removed from the Discussion section. |
10. Discussion: "LOW CHOLINESTERASE LEVELS AND GASTROINTESTINAL DISORDERS IN THE POPULATION." Introduce it correctly. Does this section need to be in uppercase? Justify the text! This also applies to "LOW CHOLINESTERASE LEVELS AND NEUROLOGICAL DISORDERS IN THE POPULATION" and other sections. |
Page 7. The subtitles were corrected by eliminating the incorrectly placed uppercase letters. |
11. Conclusions : Why didn't you address the purpose of this study? |
Pages 8 and 9. The same purpose as in the abstract is written, referring to conditions of basic services, environmental awareness, cholinesterase levels, and prevalent diseases. |
12. References: Write clearly. Pay attention to spacing and formatting according to the writing guide. |
Page 9. The references have been formatted regarding spacing, etc. |
Reviewer 2 Report
Comments and Suggestions for Authors
General comments
In your observational study, you provide interesting and very disturbing facts about the reality of fruit production in a tropical country that every consumer in the affluent countries should be aware of. Your results are certainly worth to be published. However, since this is not a "political" article to be released in a newspaper but a manuscript submitted to a scientific journal for publication, it must meet basic scientific standards which is not completely the case so far. It would be a pity if your results would remain hidden but I see no other option than to recommend a major revision, mainly amendments. I hope that my specific comments will be helpful if you try.
Specific comments
Introduction, p. 2, first para: I would like to make sure about the term "workers". To me, this is someone who is supposed to enter a crop or a plantation where pesticides have been sprayed just before to perform some agricultural task. Frequently, personal protective equipment is not worn or not available at all putting these people at high risk. In Europe at least, their function is different from that of an "operator" who is applying the pesticide and is (or should be at least) well aware of the risks and how to prevent them. Is that also your understanding or does the term have a different meaning? It might be useful to explain it.
2.2. Data analysis, first paragraph: The term "statistically significant" is a bit surprising since this can be only a result after gathering the data and their analysis. If you would just delete "statistically", it would become more clear what you mean, i.e., which factors you expect to be of concern.
Laboratory analysis of cholinesterases: First of all, I would number this paragraph (2.3; the subsequent would be 2.4 then). More important, it is not clear to me what was in fact measured: plasma or erythrocytes cholinesterases. For a toxicologist, the difference is mainly that a reduction in plasma ChE is more considered an indicator of previous exposure whereas a lower level of RBC ChE is indeed an adverse health effect in its own right. In addition, you should mention in greater detail the analytical equipment that was used for sample analysis (technical type, manufacturer).
Risk assessment and health effects: It is all correct what you have written here but, to me, this paragraph should be either part of the introduction or of the discussion. In the "Material and methods" section, you should only mention diseases/clinical entities on which informaton was obtained in the questionaire. Could be done in one sentence. The explanations should be moved either above or below.
Results, first para: What do you mean with "environmental awareness of spraying". Is it the sensory detection of sprayed pesticides in the air or did you want to know whether the participants were aware of health risks?
Table 1: What is "Social Security Instituto Ecuatoriano..."? Completely unknown to a reader abroad. Is it some sort of health insurance? Please clarify or, if not relevant for the results and conclusions, delete.
Similarly, "Energex Electrical" should be explained. Is it external supply of energy?
Table 2: You should make clear that this is all the self-estimate of the participants. The parameter "effects on maternal or live births" should be explained. I don't understand what is meant here. From the text below, I assume that the question to the participants was if they knew that sprayed pesticides might alter reproductive outcomes. You should clarify this anyway.
Table 3: Again, what ChE was measured (plasma or RBC)?
Page 6, above Table 4: Speaking about the adrenals, I would rather say: "... alter hormone syntesis or other functions.
Table 4: In the caption, should be "Low" not "low"
Table 4: One would expect all the clinical entities mentioned above (see Material and methods and my comment on that). However, "Reproductive disorder" was not listed there but appears here, without any explanation. It is not enough to come back to that in the discussion, as you did. Also, if it comes down to it, you should distinguish between women and men. (This is what you should anyway, for all analyses.)
The confidence levels must be given for the odds ratios.
Table 5: Should be deleted if you have not measured ChE (which one?) levels in all of them but provide only an estimate. Also, the increase in numbers is not that impressive.
Dicussion: I recommend to delete the last para. You must do it if you would delete Table 5. Instead, on the bottom of this part, I would address strengths and limitations of your study.
Comments on the Quality of English Language
I have spotted very few language isssues. In principle, it's fine even though a native speaker might find something more to amend. A few "Spanish elements" must be deleted from the tables (see my specific comments).
Author Response
Good afternoon, esteemed reviewers.
Thank you very much for yourIt is a pleasure to greet you. I would like to express my gratitude for the comments made on this manuscript. I would like to inform you that we have attached a document explaining the corrections made to the manuscript point by point. Additionally, comments have been added in the article itself to each paragraph based on your observations and the changes made.
attention.
Second Reviewer.
Second Observation |
Correction |
Introduction, p. 2, first paragraph: I would like to clarify the term "workers". For me, it refers to someone who is supposed to enter a crop or plantation where pesticides have been sprayed just before performing some agricultural task. Often, personal protective equipment is not used or not available at all, putting these individuals at high risk. At least in Europe, their role is different from an "operator" who is applying the pesticide and is (or should be, at least) very aware of the risks and how to prevent them. Is this also your understanding, or does the term have a different meaning? It might be helpful to clarify. |
Page 2. The term "workers" is defined in Ecuador, along with the activities they perform. |
2.2. Data analysis, first paragraph: The term "statistically significant" is a bit surprising as this can only be a result after data collection and analysis. If you simply remove "statistically", it would make clearer what you mean, that is, what factors you expect to be a cause for concern. |
Page 3. The word "statistically" is removed. |
Laboratory analysis of cholinesterases: First, I would number this paragraph (2.3; the next would then be 2.4). More importantly, it's unclear to me what was actually measured: plasma or erythrocyte cholinesterases. For a toxicologist, the difference is primarily that a reduction in plasma cholinesterase is considered more of an indicator of prior exposure, whereas a lower erythrocyte cholinesterase level is, in fact, an adverse health effect in its own right. Additionally, you should mention in more detail the analytical equipment used for sample analysis (technical type, manufacturer). |
Page 3. 2.2 The numbering is corrected. It is clarified that the measurement was of plasma cholinesterase. |
Risk assessment and health effects: Everything you have written here is correct, but for me, this paragraph should be part of the introduction or discussion. In the "Materials and Methods" section, you should only mention the diseases/clinical entities for which information was obtained in the questionnaire. This could be done in a single sentence. The explanations should go above or below. |
Page 6. A single sentence is written regarding the health risks of low cholinesterase levels. |
Results, first paragraph: What do you mean by "environmental awareness of spraying"? Is it about the sensory detection of pesticides sprayed in the air, or did you want to know if participants were aware of health risks? |
Page 4. It is explained that the goal was to know if participants were aware of the health risks of aerial spraying. |
Table 1: What is "Instituto Ecuatoriano de Seguridad Social..."? Completely unknown to a reader abroad. Is it some type of health insurance? Please clarify, or, if not relevant to the results and conclusions, remove it. Similarly, "Energex Electrical" should be explained. Is it external energy supply? |
Page 4. An explanation of the "Instituto Ecuatoriano de Seguridad Social" is provided. |
Table 2: You should clarify that all of this is the participants' self-esteem. The parameter "effects on live births or maternal effects" should be explained. I don't understand what is meant here. From the text below, I assume the question to the participants was whether they knew that sprayed pesticides could alter reproductive outcomes. Anyway, you should clarify this. |
Page 5. It is explained that the aim was to know if participants were aware of the harm caused by spraying to pregnant women and live births, and many indicated that they were aware, possibly due to their own experiences or those of a family member. |
Table 3: Again, which ChE was measured (plasma or red blood cells)? |
Page 6. It is clarified that plasma cholinesterase was measured in the blood samples in the laboratories. |
Page 6, above Table 4: Speaking of the adrenal glands, I would rather say: "... alter hormone synthesis or other functions." |
Page 6. The proposed phrase is added, "... alter hormone synthesis or other functions." |
Table 4: In the title, it should say "Low" and not "low". |
Page 6. In Table 4, the word "Low" is replaced with "low". |
Table 4: One would expect all clinical entities mentioned earlier (see Materials and Methods and my comment on that). However, "Reproductive disorder" was not on the list but appears here without explanation. It is not enough to return to it in the discussion, as you did. Also, if the time comes, you should distinguish between women and men. (This is something you should do anyway, for all analyses). Confidence levels should be given for odds ratios. |
Page 8. Regarding Table 4 and reproductive disorders, the percentage and number of men and women are presented, clarifying that these disorders affect women (e.g., miscarriages) and men (e.g., infertility and children born with disabilities). |
Table 5: It should be removed if you have not measured ChE levels (which?) in all of them but only provide an estimate. Moreover, the increase in numbers is not that impressive. |
Pages 7 and 8. Table 5 is removed, as it was a population projection based on the percentage of inhabitants, but logically, not all had their plasma cholinesterase levels measured due to resource limitations. |
Discussion: I recommend removing the last paragraph. You should do so if you eliminate Table 5. Instead, at the bottom of this section, I would address the strengths and limitations of your study. |
Pages 7 and 8. Table 5 and its interpretation are removed, and the strengths and limitations of this observational study are described. |
Round 2
Reviewer 1 Report
Comments and Suggestions for Authors
Please type the manuscript correctly, paying special attention to table spacing and writing, which can be sloppy.
Author Response
"Mr. Reviewer. Good afternoon. We appreciate your revisions and comments. Thank you very much for everything."
Reviewer 2 Report
Comments and Suggestions for Authors
General comments: I have noticed a complete revision of the manuscript resulting in major improvements. Also, you have adequately responded to the reviewer's comments and suggestions. With regard to the revised version, I have no problems with the contents any longer but still a lot of editorial comments. The main point, in my eyes, is the appropriate "placing" of certain paragraphs in the various sections. This is something that you could, perhaps, discuss and decide with the editor. I have made my proposals (that you are not supposed to follow) as you can see in my "specific comments" below.
Specific comments
Abstract: I would rather say: "do not have access to drinking water". More important, the sentence "We found that companies..." (that I completely agree with, no doubt) apears now in duplicate. Please check and delete.
Introduction: The sentence starting with "Prolonged exposure..." is correct but should be terminated after "... health risks" because the second half-sentence is redundant. A more detailed explanation is given in the following sentence and this is sufficient.
"Who have lived" should be better "who are living".
The Important para "This is truly an issue ..." is, to me, clearly a conclusion from your research or it should be at least part of the discussion. I don't understand why it is included in the introductory part already.
Also, I have a question because I am not familiar with the situation in Ecuador: Apparently, Milagro and Naranjito are separate cantons, aren't they? If so, it is suprising that you refer to "vulnerable communities" (a European reader would assume that these were mainly indiginous communities with less access to education, health and social services) with particular reference to Milagro but not to Naranjito. Is the situation there different?
Study design: In fact, I am not sure if it is necessary to report venipuncture in that much detail because you refer to this under 2.2 again. But, if you want to provide this information, be it...
I would rather use the term "plasma cholinestrase" as in the caption of 2.2 in the text, instead of "blood cholinesterase".
Results: In Table 3, "mujeres" and "hombres" should be replaced by "women" and "men".
Again, I recommend moving of an entire paragraph: To me, the para starting with "Farmers exposed to pesticides..." is a general information and statement that rather belongs to the discussion. it is not part of the "Results" section where you report what you have measured, observed, found out.
Discussion: In contrast, the entire part starting with "Low cholinsterase levels and respiratory disorders..." to "... have a planned spraying schedule", in my view, should be part of the "Results" section.
With regard to the strenghts of the study, it is surprising that you first hand look into the future, with regard to study on glyphosate in blood (which is not expected to provide meaningful results, by the way, since urine would be the much more suitable matrix). The real strength is that you could correlate a real exposure (verified by plasma cholinesterase) to real health problems. A further limitation might be the lack of measurements of erythrocyte cholinesterase that is usually considered indicative of a true toxic effect when the activity is reduced.
Conclusion: To me, the responsibility of the companies (see above) should be mentioned here.
Comments on the Quality of English Language
English, in principle, is fine. Some "remainings" in Spanish should be replaced before publication.
Author Response
Good afternoon, dear reviewer, we appreciate your observations, which are correct for the manuscript. We have made the necessary corrections and presented them in a table with page numbers, as well as highlighted them in yellow within the manuscript, with an inserted comment to facilitate your review. The manuscript only includes your observations since reviewer #1 has no further suggestions. The English translation has also been reviewed again. Thank you very much for your cooperation.
Observations from the second reviewer. |
Corrections made with page number. |
Specific comments |
|
Abstract: I would rather say: "do not have access to drinking water". More important, the sentence "We found that companies..." (that I completely agree with, no doubt) appears now in duplicate. Please check and delete. |
Pag 1.
|
Introduction: The sentence starting with "Prolonged exposure..." is correct but should be terminated after "... health risks" because the second half-sentence is redundant. A more detailed explanation is given in the following sentence and this is sufficient. |
Pag 2. The meaning of the phrase regarding prolonged exposure has been corrected |
"Who have lived" should be better "who are living". |
Pag. 2
|
The important para "This is truly an issue ..." is, to me, clearly a conclusion from your research or it should be at least part of the discussion. I don't understand why it is included in the introductory part already. |
Pag. 8. The paragraphs were moved to Discussion
|
Also, I have a question because I am not familiar with the situation in Ecuador: Apparently, Milagro and Naranjito are separate cantons, aren't they? If so, it is surprising that you refer to "vulnerable communities" (a European reader would assume that these were mainly indigenous communities with less access to education, health and social services) with particular reference to Milagro but not to Naranjito. Is the situation there different? |
Pag 2. The information regarding workers from banana agricultural areas in Milagro and Naranjito was corrected, and vulnerable communities were removed. |
Study design: In fact, I am not sure if it is necessary to report venipuncture in that much detail because you refer to this under 2.2 again. But, if you want to provide this information, be it... |
|
I would rather use the term "plasma cholinesterase" as in the caption of 2.2 in the text, instead of "blood cholinesterase". |
Pag 3.
|
Results: In Table 3, "mujeres" and "hombres" should be replaced by "women" and "men". |
Pag 5.- Men and women were included in the table.
|
Again, I recommend moving of an entire paragraph: To me, the para starting with "Farmers exposed to pesticides..." is a general information and statement that rather belongs to the discussion. it is not part of the "Results" section where you report what you have measured, observed, found out. |
Pag 8.- The paragraphs were moved to Discussion
|
Discussion: In contrast, the entire part starting with "Low cholinesterase levels and respiratory disorders..." to "... have a planned spraying schedule", in my view, should be part of the "Results" section. |
|
With regard to the strengths of the study, it is surprising that you first hand look into the future, with regard to study on glyphosate in blood (which is not expected to provide meaningful results, by the way, since urine would be the much more suitable matrix). The real strength is that you could correlate a real exposure (verified by plasma cholinesterase) to real health problems. A further limitation might be the lack of measurements of erythrocyte cholinesterase that is usually considered indicative of a true toxic effect when the activity is reduced. |
|
Conclusion: To me, the responsibility of the companies (see above) should be mentioned here. |
Pag 8. The corporate social responsibility paragraphs were unified.
|
We certify that the following article
AERIAL SPRAYING AND IMPACTS ON HUMAN HEALTH IN BANANA GROWING
AREAS OF ECUADOR.
MAURICIO GUILLEN GODOY
has undergone English language editing by MDPI. The text has been checked for correct use of grammar and common technical terms, and edited to a level suitable for reporting research in a scholarly journal. MDPI uses experienced, native English speaking editors. Full details of the editing service can be found at â–º https://www.mdpi.com/authors/english.
Basel, Switzerland
September 2024 english-83369
